# Sustainable Reciprocity Mechanism of Social Initiatives in Sport: The Mediating Effect of Gratitude

**Seung Pil Lee**

Division of International Sport and Leisure, Hankuk University of Foreign Studies, Gyunggi-do 17035, Korea; seungpil@hufs.ac.kr; Tel.: +82-31-330-4719

**Abstract:** The study aims to explore a conceptual model for the sustainable reciprocity relationships in sport-based initiatives and empirically test the model and its underlying mechanism in the context of a real sport-based national initiative. Adapting a seminal work from social work literature as a theoretical framework and the following measurement for the social impact of sport from sport management literature, a conceptual model addressing sport participation, gratitude, social benefits, and prosocial behaviors is presented. Two separate surveys were conducted through face-to-face interviews with independent and random samples representing the Singaporean residents in October 2014 for Study 1 (n = 500) and February 2015 for Study 2 (n = 501). The results demonstrate that the frequency of participation in a range of daily sport activities of a national sport initiative positively influences the perceived value of social capital and health literacy through the mediation effect of gratitude. The study also demonstrates that participation in sport activities positively influences prosocial behavioral intention through the serial mediation effect of gratitude and social capital. The findings implies how we better understand and utilize the dynamic power of gratitude to sustain the win–win relationships to multi-stakeholders in the contexts of sport-based initiatives based on the nature of reciprocity.

**Keywords:** social capital; health literacy; prosocial behavior; corporate social responsibility; sponsorship

## 1. Introduction

The number of organizations participating in sport-based social initiatives in the forms of various collaborations such as sponsorship, partnerships, and corporate social responsibility (CSR) is continually growing. Additionally, sports are the fastest growing sponsorship categories with causes [1]. Many researchers have investigated the effectiveness of those collaborative initiatives for organizational goals [2–5]. There has been, however, little research to investigate the actual benefits for participants or beneficiaries, despite the fact that those social initiatives in sport are literally beneficiary-oriented. One of the challenges might be that the social outcomes for beneficiaries from sport-based social initiatives frequently are "difficult-to-measure or intangible" constructs in areas related to health, well-being, social capital, societal equity, education, gender equality, sustainability, and peace [6]. Subsequently, the potential might be undervalued and underleveraged to stakeholders (e.g., general public, athletes, volunteers/donors, corporate sponsors, employees, nonprofits, government) [6]. For this reason, consistent support and investments and the presumed win–win relationships for multi-stakeholders in sport-based social initiatives may be limited. Meanwhile, it is one thing for a program to do good work, and it is another thing to measure and communicate these successful mechanisms, relationships, and outcomes to stakeholders in a sport-based social initiative. pecially, measuring and communicating the invisible outcomes to various stakeholders from sport-initiatives would be crucial in order to sustain the win–win relationships. That is, the measurement of outcomes for participants/beneficiaries in sport-based social initiatives could be useful and leveraged to attract

social, political, and financial support in the various forms such as corporate sponsorship, donations, and volunteering [6]. For example, according to the work of Lee and Babiak [3], communicating measured societal value can be beneficial to both nonprofits and corporate sponsors in CSR partnership initiatives by increased donation intentions and reduced perception of corporate hypocrisy because the standardized measurement can effectively validate the invisible societal functional values of social initiatives with the general public [3].

In sport-based social initiatives of social sponsorship or CSR practices, we particularly note that participants/beneficiaries can experience or receive either tangible or intangible positive social benefits provided from other entities. Considering these characteristics, gratitude might be regarded as an underlying factor to influence the social benefits for participants/beneficiaries in sport-based social sponsorship initiatives and CSR practices. Additionally, the perception of gratitude might play a role in motivating participants, benefactors (e.g., corporate sponsors, volunteers, donors, NPO partners), and the general public to act more prosocially. However, there has been little research to investigate gratitude as an important construct to influence the perceived social outcomes related to social capital, well-being, and health for participants in cause-oriented sponsorship and CSR literature. Additionally, there has been insufficient research to investigate how gratitude can better play a role in motivating beneficiaries, benefactors (e.g., corporate sponsors, volunteers, donors, NPO partners), and the general public to act more prosocially in collaborative social initiatives, ultimately promoting gratitude practices in organizational settings, communities, and society at large in sport-based social sponsorship and CSR practices.

In summary, when we intend to accomplish and sustain the so-called win–win relationships with multi-stakeholders through sport in the real business environment, it would be an efficient starting point to develop a conceptual model to deal with the measurement issue and the gratitude-based reciprocity relationship, as we mentioned above. Thus, the first objective of the study is to explore a conceptual model for the sustainable reciprocity relationships among sport participation, gratitude, social benefits, and prosocial behaviors in a sport-based initiative based on previous literature on measurements for the social impact of sport and gratitude. The second objective is to empirically test the conceptual model and its underlying mechanism of structural relationships among sport, gratitude, perceived social outcomes for participants, and their prosocial behaviors in the context of a real sport-based national initiative, "Vision 2030: Live Better through Sport" in Singapore, applying a developed instrument to measure the social impact of sport [6]. Thus, the study aims to better understand the sustainable reciprocity mechanism of sport-based social initiatives, focusing on measurement of societal value and gratitude in sport.

## 2. Theoretical Foundation and Hypotheses Development

### 2.1. Sport and Social Benefits

As one of the social elements in society, sport is popularly recognized as a tool to create positive social value by generating social capital central to social inclusion, empowerment, well-being, community development, health education, and youth development [7–20]. Previous research supports this argument in various contexts such as the UK, Canada, and the USA [16,21–24]. According to the work of Lee, Cornwell, and Babiak [6], there has been consistent support for the potential of sport's contribution to society, but little empirical evidence in a standardized and systematic format. In addressing this issue, first, they chose five core areas to which sport can make a contribution to society in terms of social capital, collective identities, health literacy, well-being, and human capital by adapting the conceptual work of Lawson [11] and other scholars' supporting literature [25,26].

According to the work of Lawson [11], sport, exercise, and physical education (SEPE) can develop and enhance social networks among participants, their families, residents of the community, and professions, generating social trust and norms of civil society. Secondly, he argues that SEPE can be designed to contribute to the development of collective identities by linking intergroup differences,

promoting solidarity and social integration. Third, Lawson conceptualizes that SEPE can contribute to health-enhancing environments. Further, he argued that SEPE can improve well-being and health, nurture balanced relationships, and offer opportunities for identity development. Finally, he argues that SEPE can make a contribution to human capital development related to knowledge, skills, attitude, competence, capacity, and citizenship of individuals and groups. Following this foundation for the social contribution of sport, Lee, Cornwell, and Babiak [6] developed a measurement for the chosen five core constructs to be influenced by sport based on extensive literature review of past measures in the respective areas. Additionally, they provided their paraphrased definitions based on supporting literature (see Table 1). The findings demonstrated that a structural equation model based on a two group comparison by the awareness of a major charity sport event in a community reveals that the frequencies of exposures to community-oriented sports (e.g., intramural sport, local softball league, local tennis tournament) and participation in individual recreation sports positively influences the development of social capital, collective identities, and health literacy [6]. In addition, they found that the awareness of a major charity sport event in the community played a moderating role in these causal relationships [6]. Previous research also argues that the general public's participation in sports activities in daily life could be a fundamental factor to influence various developmental outcomes in the context of grassroots sports [27,28].

**Table 1.** Definitions and sources for conceptual measures of the social impact of sport [6].

| Construct | Definition | Sources |
|---|---|---|
| Social capital | Social relationships and conditions including trustworthy and diverse networks, social proactivity, and participation in community are conducive to cooperation for mutual success in society | [11,24,29–33] |
| Collective identities | The sense of belonging to a social group or community reflecting self-categorization with positive attitude and important self-concept in a social context | [11,34–36] |
| Health literacy | An individual's functional, interactive, and critical abilities to understand and use healthcare information to make appropriate health decisions | [11,37–39] |
| Well-being | Harmonious life quality in both psychological and economic aspects for human function and development | [11,40,41] |
| Human capital | The attributes of individuals in terms of knowledge, skills, competencies, and attitudes conducive to personal development and societal well-being | [11,42,43] |

Adapting Lawson's [11] seminal work as a theoretical framework and the following conceptual measurement model and the empirical evidence of Lee, Cornwell, and Babiak [6], we hypothesize that frequency of participation in a range of activities of sport-based social initiative are expected to influence the multi-dimensional values of social benefits related to social relationships and development, well-being, human capital, and health. (see Figure 1).

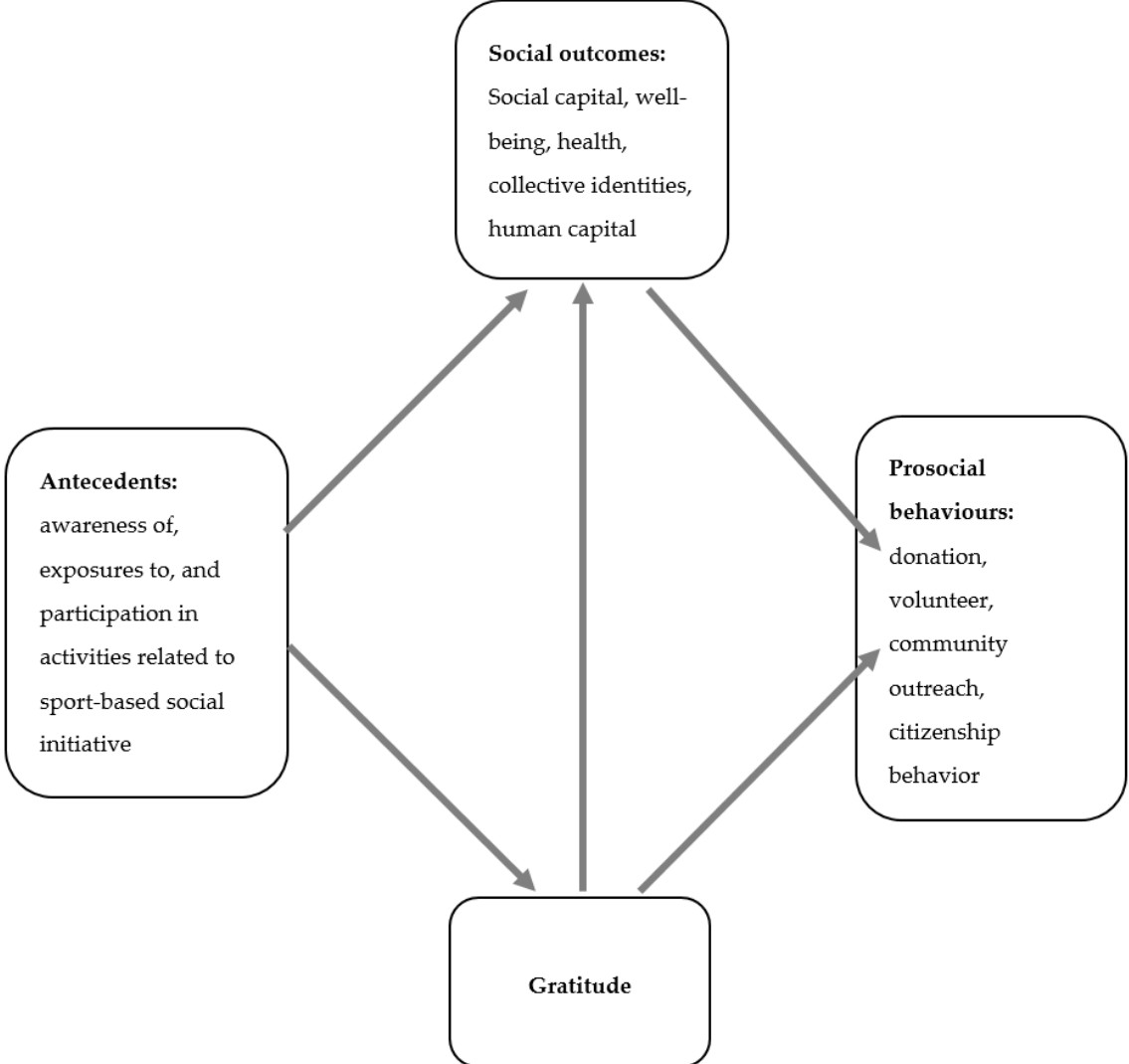

**Figure 1.** A conceptual model of the reciprocity mechanism in sport-based social initiative.

**Hypothesis 1 (H1).** *The frequency of participation in a range of sport activities positively influences the perceived social benefits of social capital, collective identities, health literacy, well-being, and human capital of participants in a sport-based initiative.*

*2.2. Gratitude, Social Outcomes, and Prosocial Behaviors*

In sport-based social initiatives in the forms of social sponsorship or CSR practices, participants experience or receive positive benefits either tangibly or intangibly offered from other individuals or organizations. For example, as a government's administrative sports organization, the Korea Sport Promotion Foundation (KSPO) launched a "sports voucher" program in 2009 to partially subsidize lectures or entrance fees to sports classes or events for children and youth from low-income families as one of the channels of promoting healthy and active society and nation [44]. More than 140,000 children and youth participated in the sport voucher program for the last six years from 2009 to 2014, receiving benefits from the initiative.

Given these characteristics, gratitude, defined as "a sense of thankfulness and joy in response to receiving a gift, whether the gift be a tangible benefit from a specific other or a moment of peaceful bliss evoked by natural beauty" [45] (p. 554), might be a fundamental construct to influence the social outcomes for participants or beneficiaries in sport-based social initiatives. In fact, many running organizations introduce several reasons to be thankful for running [46,47]. For example, they reason

that running offers opportunities to get ourselves outside of the house or work and experience the natural beauty of creation by seeing others, animals, trees, and the world on a run. Therefore, gratitude might be a more immediate outcome for participants/beneficiaries to enjoy from fun and pleasant experiences in sport activities, especially cause-oriented sport activities, given that gratitude also can be referred to as pleasant state and is linked with positive emotions including contentment [48].

There has been consistent support for the positive correlation between gratitude and positive outcomes related well-being, health, social relationships, and youth development. For example, Haidt [49] argued gratitude promotes benefits exchanges and societal well-being [49]. Emmons and McCullough [50] argued that gratitude is strongly correlated with healthy psychological and social functions focusing on self-improvement and social ties [50]. Wood et al. also summarized that gratitude is associated with positive emotional functioning, lower dysfunction, and positive social relationships [51]. Given this theoretical foundation in previous research, we hypothesize that the gratitude of participants/beneficiaries is positively correlated with the social outcomes in the context of sport-based social initiatives. Specifically, we hypothesize that the perceived gratitude of participants mediates the relationship from their participation in sport activities to their social benefits in sport-based social initiatives (see Figure 1).

**Hypothesis 2 (H2).** *The frequency of participation in a range of sport activities related to a sport-based initiative positively influences the perceived social benefits through the mediation of gratitude.*

In addition, given that gratitude would be a fundamental construct to influence the relationships and outcomes related to cause-oriented sport sponsorship and CSR practices, we want to view it as a more integrated mechanism. Notably, McCullough et al. [52] originally proposed the three functions of gratitude as (1) "a moral barometer for beneficiaries by signaling the value of the relationship with benefactor for the gift bestowed upon them, (2) a moral reinforcer by increasing the probability that the benefactor will bestow gifts again in the future, and (3) a moral motive by spurring beneficiaries to respond prosocially toward the benefactor or other people" [53] (p. 312). Recent experimental studies offered convincing evidence in support of the moral motive function of gratitude [54–56], which spurs beneficiaries to respond prosocially toward other people as well as the benefactor. This dynamic nature of gratitude can also play a key role in encouraging participants, benefactors, and the general public to act more prosocially in the context of a sport-based social initiative. Thus, we raise an exploratory question: whether gratitude of participants can mediate the relationship from participation in sport and their prosocial behaviors in sport-based social initiatives (see Figure 1).

**Hypothesis 3 (H3).** *The frequency of participation in a range of sport activities related to a sport-based initiative positively influences their prosocial behavior through the mediation of gratitude.*

**Hypothesis 4 (H4).** *The frequency of participation in a range of sport activities related to a sport-based initiative positively influences their prosocial behavior through the serial mediation of gratitude and their perceived social benefits.*

In the following empirical work, Study 1 and Study 2 were designed to examine the relationships between sport participation, gratitude, social benefits, and the intention of prosocial behavior in a sport-based initiative. Two separate surveys were administered through face-to-face interviews with independent and random samples representative of the Singaporean residents in terms of gender, race, income, education, and age (20–59) in October 2014 for Study 1 and February 2015 for Study 2. We tested the hypothesized conceptual relationships in the context of a national sport-based social initiative of Singapore. The "Vision 2030: Live Better through Sport" (a short name is used subsequently as Vision 2030) is a sport initiative led by a sport government organization, Sport Singapore, with active participation from the general public and private sectors in Singapore. As one of the fundamental engines for it, "Active Singapore" was launched in April 2014 in order to create a sporting ecosystem

with sports programs available, accessible, and affordable to everyone regardless of their skill level and age. Therefore, it offers a specific target population in the context of a sport-based development to examine the actual developmental relationships and outcomes with realities and complexities.

## 3. Study 1

### 3.1. Method

#### 3.1.1. Participants and Procedures

A random sample (n = 500) was obtained from the Department of Statistics Singapore at a charge (see Table 2). If an eligible respondent was not available for interview, he/she was replaced by the next household with the "nearest door". This procedure was followed until all the interviews were conducted. Five one-dollar vouchers (totaling S$5) were given to each respondent as a token of compensation and appreciation for participation. The author also procured a local survey company with 20 trained interviewers for door-to-door survey administration for three weeks. Before starting the fieldwork, all interviewers went through an intensive training in a briefing session designed to acquaint them with the sampling procedures and the importance of these procedures. A briefing session for all interviewers and supervisors explained the survey objectives, interviewing procedures, and how the questionnaire should be administered. Mock interviews among interviewers also ensured that they have achieved a thorough understanding of how the questionnaire should be administered.

**Table 2.** Demographics of a sample representative of the Singaporean residents (Study 1).

| Sample Size | | % | Number of Respondents |
|---|---|---|---|
| Age Group | 20–29 | 25.6 | 128 |
| | 30–39 | 26.8 | 134 |
| | 40–49 | 27 | 135 |
| | 50–59 | 20.6 | 103 |
| Race | Chinese | 66.4 | 332 |
| | Malay | 18 | 90 |
| | Indian | 12.6 | 63 |
| | Others | 3 | 15 |
| Sex | Male | 42.4 | 212 |
| | Female | 57.6 | 288 |
| Monthly Household Income | S$1999 and below | 9.6 | 48 |
| | S$2000–S$3999 | 22 | 110 |
| | S$4000–S$5999 | 28.4 | 142 |
| | S$6000–S$8999 | 17.4 | 87 |
| | S$9000 and above | 15.8 | 79 |
| | Prefer not to disclose | 6.8 | 34 |
| Education Level | Below Secondary | 5.4 | 27 |
| | Secondary | 26.8 | 134 |
| | Post-Secondary (Non-Tertiary) | 13.8 | 69 |
| | Diploma and Professional Qualification | 27.6 | 138 |
| | University | 26.2 | 131 |
| | Others | 0.2 | 1 |
| TOTAL | | 100 | 500 |

#### 3.1.2. Measures

Social Outcomes

Appling the newly-developed measurement of Lee, Cornwell and Babiak [6], a total of 23 items were used to assess the social outcomes of social capital (5 items), well-being (5 items), health literacy (4 items), human capital (5 items), and collective identity (4 items). They were measured by a 10-point Likert Scale (1 = strongly disagree, 10 = strongly agree). See Table 3 for this instrument.

**Table 3.** Measures of Social Outcomes and Gratitude: Factor Loadings ($\beta$), Cronbach's Alpha ($\alpha$), and Average Variance Extracted (AVE).

| Factor and Item | B | $\alpha$ | AVE |
|---|---|---|---|
| **Social capital** [6] | | | |
| GS-1. I have trustworthy social interaction and cooperation in daily activities with the people in my community. | 0.72 | | |
| GS-2. I currently enjoy trustworthy interaction and cooperation with the people in my community. | 0.75 | | |
| GS-3. Generally, I trust and cooperate with people in my social networks. | 0.65 | 0.88 | 0.52 |
| GS-4. When I interact with people in my community, I feel a common sense of trust and cooperation. | 0.75 | | |
| GS-5. I feel I work with trustworthy and cooperative people in my community. | 0.72 | | |
| **Well-being** [6] | | | |
| GW-1. I feel I am continually growing and developing as a person. | 0.74 | | |
| GW-2. I feel good about my whole life. | 0.73 | 0.87 | 0.55 |
| GW-3. I generally feel healthy, happy, and appreciated. | 0.75 | | |
| GW-4. I feel confident in my ability to handle most things in my life. | 0.74 | | |
| **Health literacy** [6] | | | |
| GH-1. I have a basic understanding and communication skills needed to maintain my health | 0.72 | | |
| GH-2. I currently enjoy trustworthy interaction and cooperation with the people in my community. | 0.77 | 0.90 | 0.57 |
| GH-3. I have the capability to obtain, understand, and process basic health information and services to make appropriate health decisions. | 0.76 | | |
| GH-4. I understand I am in control of my health. | 0.76 | | |
| **Collective identities** [6] | | | |
| GC-1. I have a strong sense of belonging to the community or group where I live or work. | 0.76 | | |
| GC-2. I have a shared feeling of "we" or "groupness" with the people in my community or group where I live or work. | 0.82 | 0.76 | 0.54 |
| GC-3. I have shared goals, ideas, or opinions with the people in my community or group where I live or work. | 0.68 | | |
| GC-4. I have similar goals, ideas, or views to the people in my community or group where I live or work. | 0.68 | | |
| **Human capital** [6] | | | |
| GHC-1. I feel I am continually growing and developing as a person. | 0.71 | | |
| GHC-2. I have opportunities to continue developing knowledge, skills, and competencies. | 0.72 | | |
| GHC-3. I have the necessary knowledge, skills, and competence to develop as a person. | 0.83 | 0.76 | 0.54 |
| GHC-4. I am continually making efforts to improve my social and economic well-being. | 0.80 | | |
| GHC-5. I am committed to improve my social and economic well-being. | 0.77 | | |
| **Gratitude** [57,58] | | | |
| GR-1. I have so much in life to be thankful for. | 0.90 | | |
| GR-2. I am thankful to a wide variety of people. | 0.96 | 0.84 | 0.59 |
| GR-3. If I had to list everything that I felt thankful for, it would be a very long list. | 0.77 | | |

Gratitude

Adapting the measurements of Thomas and Watkins [57] and McCullough, Emmons, and Tsang [58], three items were adapted to measure gratitude [57,58]. They were measured by a 10-point Likert Scale (1 = strongly disagree, 10 = strongly agree). See Table 3 for the instrument.

Prosocial Behaviors

As one of the intentions of prosocial behaviors related to the national sport-based development initiative of Singapore, donation behavior intention was measured, adapting the donation mechanism of contingent valuation method (CVM). The respondents were exposed to a contingent scenario, which elicits willingness to donate (WTD) to the sport-based societal foundation. It is the National Football Academy Foundation, hypothetically established as a non-profit to promote the societal values of civic pride, community bonds, national pride and identity, inspiration, and role modeling of Singapore's young footballers. We use the multiple bounded discrete choice (MBDC) elicitation developed by Welsh and Poe [59] to reduce the hypothetical bias of contingent valuation [59]. It requires respondents to express their donation decision certainty for their amount of willingness to donate (WTD) as one of the five choices including definitely yes, probably yes, don't know, probably no, and definitely no. Individuals' minimum amount with "definitely yes" is retained as the dependent variable of WTD for each respondent. In this way, we can more conservatively measures their donation intention by reducing the hypothetical bias found in donation format of CVM (see Appendix A).

Independent Variables

Frequencies of participation in sports activities (e.g., jogging, swimming, cycling, martial arts, soccer, basketball, volleyball, tennis, golf, cricket) were measured by the format: never (0), once every

quarter of a year (0.08 per week), once a month (0.25 per week), once every other week (0.5 per week), 1 time a week (1 per week), 2–3 times a week (2.5 per week), 4–5 times a week (4.5 per week), or 6–7 times a week (6.5 per week).

*3.2. Results*

3.2.1. Measurement Model

We employed confirmatory factor analysis to assess the validity of the multiple items measuring the constructs of social capital, well-being, health literacy, collective identities and human capital, and gratitude. Using the maximum likelihood estimation procedure, the measurement model represents the six constructs with a good fit to the data (N = 500, Chi-Square/df = 982.108/231 = 4.252, CFI = 0.929, TLI = 0.908, IFI = 0.929, RFI = 0.882, NFI = 0.910, RMSEA = 0.081). All indicators loaded significantly on the constructs as expected, showing the convergent validity. They ranged from 0.65 to 0.75 for social capital, 0.73 to 0.76 for well-being, 0.72 to 0.77 for health literacy, 0.68 to 0.82 for collective identities, 0.71 to 0.83 for human capital, and 0.77 to 0.90 for gratitude. Additionally, all the constructs also exhibited reliability with Cronbach's alpha ranging from 0.84 to 0.90 (see Table 3). In addition, there is no correlation more than 0.85 between constructs of social capital, health literacy, and gratitude [60], and there is no squared correlation between one and any others bigger than AVE for each construct for social capital, health literacy, and gratitude [61] (Fornell and Larcker, 1981) (see Table 4). Thus, discriminant validity is established for social capital, health literacy, and gratitude. However, due to strong correlations (0.99 for social capital and collective identities, 0.96 for health literacy and human capital, 0.97 for health literacy and well-being), discriminant validity is not established for these constructs of human capital, collective identities, and well-being. In other words, social capital was captured as a similar construct to collective identities, and health literacy was captured a similar construct to well-being and human capital from the respondents of the general public in Singapore. We further discuss this measurement challenge in the discussion section. In the following examination for structural relationships, therefore, we focus on two separate dimensional constructs of social capital and health literacy as social benefits from the sport-based development initiative of Vision 2030.

**Table 4.** Correlations among the Constructs of Social Outcomes and Gratitude.

| | 1 | 2 | 3 | 4 | 5 | 6 |
|---|---|---|---|---|---|---|
| 1. Social capital | 1.00 | | | | | |
| 2. Collective identity | 0.99 | 1.00 | | | | |
| 3. Health literacy | 0.39 | 0.66 | 1.00 | | | |
| 4. Well-being | 0.52 | 0.40 | 0.97 | 1.00 | | |
| 5. Human capital | 0.56 | 0.28 | 0.96 | 0.96 | 1.00 | |
| 6. Gratitude | 0.39 | 0.28 | 0.37 | 0.37 | 0.40 | 1.00 |

3.2.2. Structural Model

Structural equation modeling (SEM) was used to examine the social outcomes in hypothesized directions and examine any underlying mechanism of structural relationships among sport participation, gratitude, the perceived social outcomes for beneficiaries, and their prosocial behavior in sport-based social initiatives [62]. Figure 2 depicts the serial mediation roles of gratitude and perceived social capital in the structural relationships from the actual participation in a range of sport activities and the relevant prosocial behaviors of Vision 2030 (n = 500, Chi-Square/DF = 281.897/66 = 4.271, CFI =0.944, TLI = 0.923, IFI = 0.946, RFI = 0.902, NFI =0.929, RMSEA = 0.081).

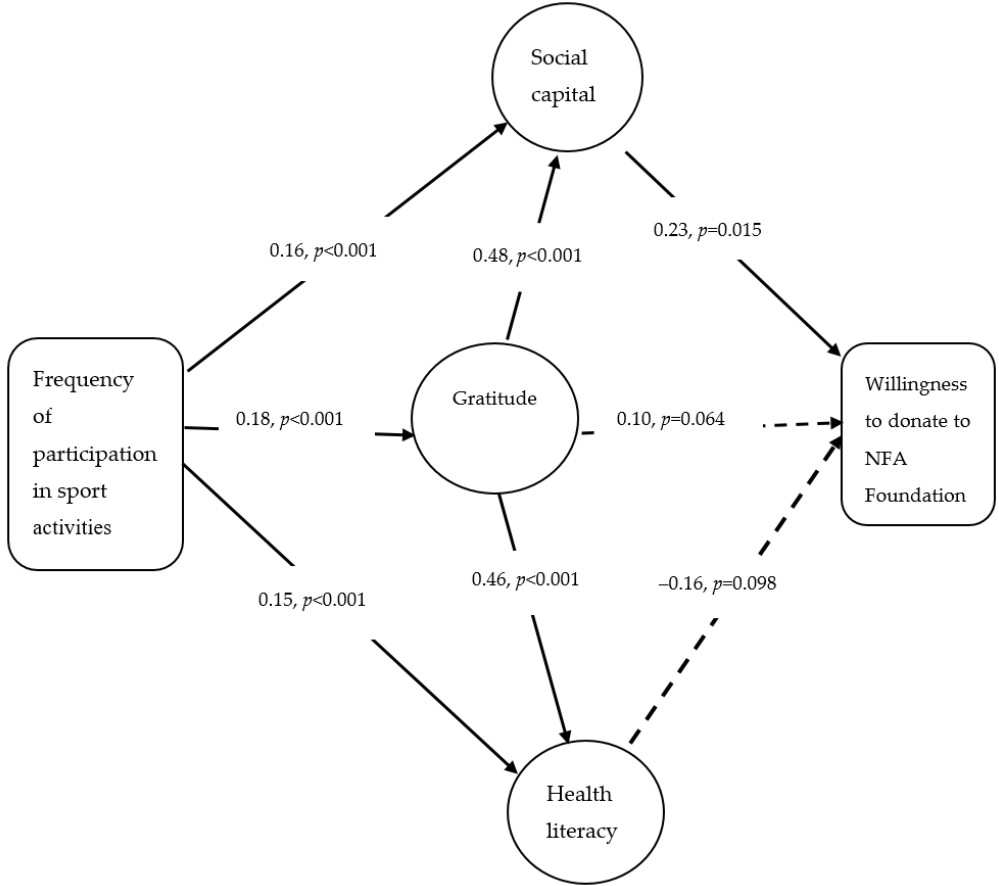

**Figure 2.** Structural relationships among gratitude, social outcomes, and prosocial behaviors in sport-based social initiative (Study 1: n = 500, Chi-Square/DF = 281.897/66 = 4.271, CFI = 0.944, TLI = 0.923, IFI = 0.946, RFI = 0.902, NFI = 0.929, RMSEA = 0.081).

### 3.3. Findings

As shown in the model (Figure 2), frequency of participation in a range of sport activities positively influences social capital ($\beta$ = 0.16, $p$ < 0.001) and health literacy ($\beta$ = 0.15, $p$ < 0.001) directly, supporting Hypothesis 1. It also shows that frequency of participation in a range of sport activities positively influences the gratitude of beneficiaries/participants ($\beta$ = 0.18, $p$ < 0.001), which positively influences their social outcomes of social capital ($\beta$ = 0.48, $p$ < 0.001) and health literacy ($\beta$ = 0.46, $p$ < 0.001), supporting Hypothesis 2. On the other hand, it does not show that the frequency of participation in a range of sport activities influences their prosocial behavior intention of willingness to donate (WTD) to sport-based societal foundation through the mediation of gratitude ($\beta$ = 0.10, $p$ = 0.064), which does not support Hypothesis 3. It shows, however, that the frequency of participation in a range of sport activities related to a sport-based initiative positively influences their prosocial behavior of willingness to donate (WTD) through the serial mediation of gratitude and their perceived social capital ($\beta$ = 0.23, $p$ = 0.015), supporting Hypothesis 4. Interestingly, health literacy as one of the social benefits is not a significant mediating factor to influence the prosocial behavior intention of donation ($\beta$ = −0.16, $p$ = 0.098). The findings are, however, limited to be generalizable.

## 4. Study 2

### 4.1. Method

To replicate the findings, we conducted Study 2 with a different sample four months later in February 2015 (see Table 5). Another random sample (n = 501) was obtained from the Department of

Statistics Singapore at a charge. The same approach as Study 1 was used for survey administration in Study 2 in the context of sport-based social initiative of Vision 2030.

**Table 5.** Demographics of a sample representative of the Singaporean residents (Study 2).

| Sample Size | | % | Number of Respondents |
|---|---|---|---|
| | 20–29 | 27.5 | 138 |
| Age Group | 30–39 | 22.4 | 112 |
| | 40–49 | 25.2 | 126 |
| | 50–59 | 25 | 125 |
| | Chinese | 72.9 | 365 |
| Race | Malay | 15.2 | 76 |
| | Indian | 10.2 | 51 |
| | Others | 1.7 | 9 |
| Sex | Male | 49.1 | 246 |
| | Female | 50.9 | 255 |
| | S$1999 and below | 12.8 | 64 |
| | S$2000–S$3999 | 23 | 115 |
| Monthly Household Income | S$4000–S$5999 | 14.8 | 74 |
| | S$6000–S$8999 | 11.2 | 56 |
| | S$9000 and above | 11.2 | 56 |
| | Prefer not to disclose | 27.1 | 136 |
| | Below Secondary | 5 | 25 |
| | Secondary | 26.3 | 132 |
| Education Level | Post-Secondary (Non-Tertiary) | 9.4 | 47 |
| | Diploma and Professional Qualification | 27.5 | 138 |
| | University | 31.7 | 159 |
| | Others | - | - |
| TOTAL | | 100 | 501 |

Measures

The same donation mechanism of CVM was used to measure willingness to donate (WTD) as in Study 1. Likewise, social outcomes and gratitude were measured in the same items by a 10 point-Likert Scale (1 = strongly disagree, 10 = strongly agree). Additionally, independent variables were composed of the same format as Study 1: never (0), once every quarter of a year (0.08 per week), once a month (0.25 per week), once every other week (0.5 per week), 1 time a week (1 per week), 2–3 times a week (2.5 per week), 4–5 times a week (4.5 per week), or 6–7 times a week (6.5 per week).

*4.2. Results*

4.2.1. Measurement Model

We employed a confirmatory factor analysis to assess the validity of multiple items measuring the constructs of social capital, collective identities, health literacy, well-being, human capital, and gratitude. Using the maximum likelihood estimation procedure, the measurement model represents the six constructs with a good fit to the data (N = 501, Chi-Square/df = 660.589/136 = 4.857, CFI = 0.922, TLI = 0.902, IFI = 0.923, RFI = 0.880, NFI = 0.905, RMSEA = 0.088). All indicators loaded significantly on the constructs as expected, showing the convergent validity. They ranged from 0.72 to 0.78 for social capital, 0.72 to 0.76 for collective identities, 0.75 to 0.81 health literacy, 0.77 to 0.81 for well-being, 0.69 to 0.82 for human capital, and 0.86 to 92 for gratitude. In addition, all the constructs showed reliability, with Cronbach's alphas ranging from 0.85 to 0.92. However, as we found in Study 1, discriminant validity is not established for the constructs of human capital, collective identities, or well-being due to strong correlations again (0.99 for social capital and collective identities, 0.93 for health literacy and human capital, 0.87 for health literacy and well-being). In the following examination for structural relationships, therefore, we focus on two separate dimensional constructs of social capital and health

literacy as social benefits from the sport-based development initiative of Vision 2030. We further discuss this measurement issue in the discussion section.

### 4.2.2. Structural Model

Structural equation modeling (SEM) was used to test the hypotheses, including mediation effects in the causal relationship [58]. Figure 3 depicts the mediation roles of gratitude and social benefits in the relationships between participation in sport activities and donation intention of prosocial behavior (n = 501, Chi-Square/DF = 194.591/66 = 2.948, CFI = 0.965, TLI = 0.952, IFI = 0.965, RFI = 0.922, NFI = 0.948, RMSEA = 0.061). As shown in the model in Figure 3, frequency of participation in a range of sport activities positively influences social capital ($\beta$ = 0.09, $p$ = 0.032) and health literacy ($\beta$ = 0.14, $p$ < 0.001) directly, supporting Hypothesis 1 again. Additionally, participation in sport activities significantly enhanced gratitude ($\beta$ = 0.16, $p$ < 0.001), which subsequently enhanced social capital ($\beta$ = 0.42, $p$ < 0.001) and health literacy ($\beta$ = 0.50, $p$ < 0.001), supporting H2 again. The model also replicates the finding that participation in sport activities enhanced donation intention of prosocial behaviors through the serial mediation effects of gratitude and social capital, supporting H4. Interestingly, the direct effect of gratitude on donation intention of prosocial behavior was not significant ($\beta$ = 0.09, $p$ = 0.108), and the direct effect of health literacy on donation intention of prosocial behavior was also not significant ($\beta$ = −0.10, $p$ = 0.166).

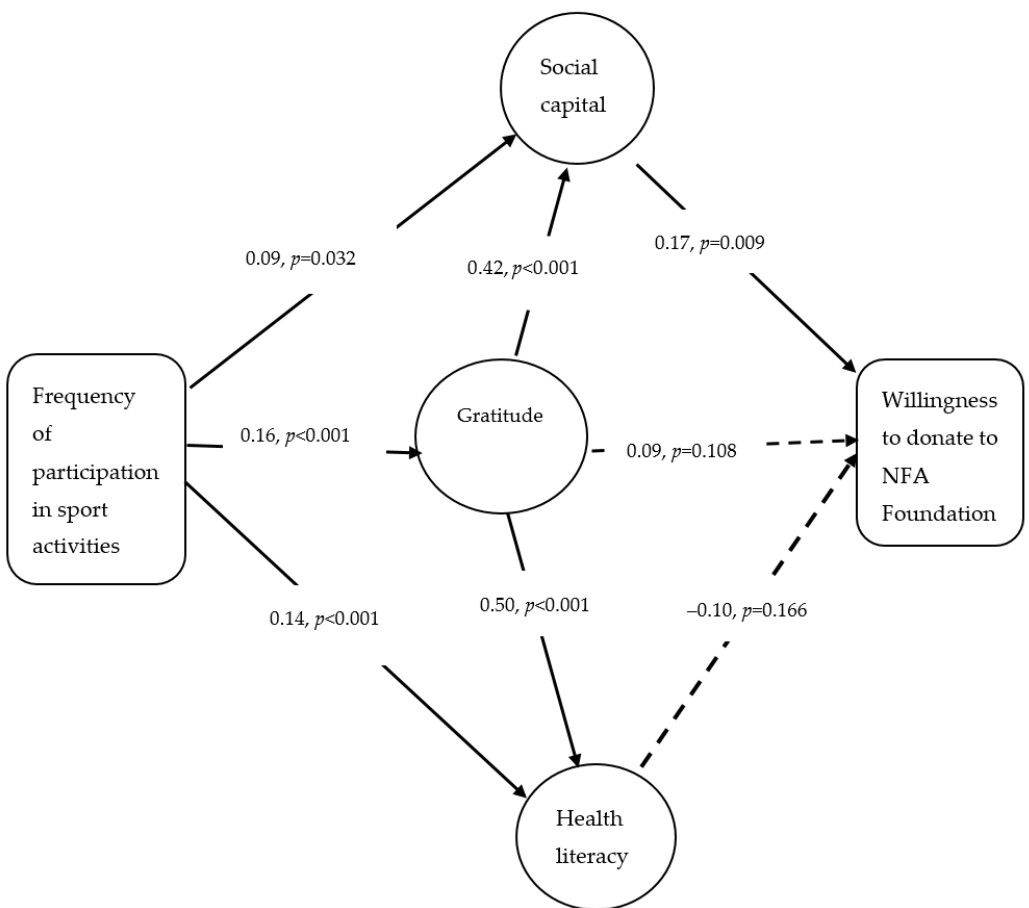

**Figure 3.** Structural relationships among gratitude, social outcomes, and prosocial behaviors in sport-based social initiative (Study 2: n = 501, Chi-Square/DF = 194.591/66 = 2.948, CFI = 0.965, TLI = 0.952, IFI = 0.965, RFI = 0.922, NFI = 0.948, RMSEA = 0.061).

### 4.3. Findings

Study 2 replicated the findings that participation in sport activities enhances donation intention of prosocial behavior in a sport-based social initiative with another representative sample. Importantly, the work demonstrates the serial mediation roles of gratitude and social capital that link participation in sport activities to donation intention of prosocial behavior. As a complementary effort to reinforce the reciprocity relationship in sport-based social initiatives, the findings showed how gratitude of participants can play a significant role in enhancing prosocial behaviors in sport-based social initiatives. The study also identified the underlying needs of social capital for those exposed to sport-based social initiatives in enhancing their prosocial behaviors.

## 5. Discussion

An important contribution of this study is to propose a conceptual model of a sustainable reciprocity mechanism in sport-based social initiatives by integrating the standardized measurement constructs of social outcomes and gratitude as well as the relevant prosocial behaviors from interdisciplinary literature of sport management, social work, and psychology. Additionally, applying the conceptual model into a sport-based national initiative with a sample representative of national population, the study empirically shows that this reciprocity mechanism can materialize in a real sport-based development initiative such as in Singapore. Additionally, the empirical structural model visualizes the mediation effect of gratitude in the reciprocity mechanism, which significantly influences the two dimensions of social outcomes: social capital and health literacy for beneficiaries/participants. When we remove the construct of gratitude in the structural equation model, the explanatory power ($r^2$) for the social outcomes becomes significantly reduced from 0.28 to 0.06 for social capital and from 0.26 to 0.05 for health literacy in Study 1. As for Study 2, the explanatory power ($r^2$) for the social outcomes becomes significantly reduced from 0.20 to 0.03 for social capital and from 0.26 to 0.05 for health literacy. It confirms the significance of the mediating role that gratitude is playing in the structural relationships of the reciprocity mechanism in sport-based social initiatives.

Additionally, the current study provides a potential regarding how sport-based social initiatives should be designed, managed, and developed as a self-reinforcing or sustainable system based on the reciprocity nature. Previous research showed significant positive relationships between gratitude and athletes' well-being and team satisfaction in Taiwan [63]. Additionally, researchers from sport psychology found that adolescent athletes' gratitude positively influences life satisfaction through the mediation of perceived team cohesion [64]. Further, many researchers demonstrate that gratitude influences prosocial behaviors [54–56,64]. However, these relationships from previous works are still fragmentary and do not fully support a sustainable system of sport-based social initiatives in a holistic approach. In this challenge, the proposed conceptual model is more necessary and meaningful to seek the sustaining win–win relationships of sport-based social initiatives. Especially, it implies how to deal with the measurement issue of intangible societal value in sports and leverage gratitude as another mediating factor to facilitate the more successful outcomes in sport-based social initiatives. In addition, while previous research found that gratitude can lead to prosocial behavior [54–56,64], the current research shows the serial mediation effects of gratitude and social capital to motivate prosocial behaviors in sport-based social initiatives. It implies how the measurement and communication of social capital-related values in sport is significant to influence prosocial behaviors in social initiatives. On the one hand, we found that health literacy as one of social outcomes is not a significant mediator to influence prosocial behavior in the model. One of the reasons for it might be that health literacy indicates more personal outcomes than social benefits and that this individualized nature of health literacy might result in the no significant relationship with prosocial behavior. Additionally, further research is required to examine this relationship in a longitudinal way, given that health literacy is a more immediate outcome than health and social health.

Especially noteworthy is the fact that the empirical model shows that frequency of participation in a range of sport activities in daily life can positively influence gratitude of participants/beneficiaries.

Given that there has been no research examining the relationship between sport participation and gratitude, we examined this structural relationship in an exploratory question. This is the first study to empirically examine that participation in recreational sport activities in daily life can positively influence gratitude. The findings offer significant implications to both academics and practitioners in sport management and industries regarding how to better understand and leverage the dynamic power of gratitude in a sport-based social initiative. Future research is required to develop a stronger and deeper theoretical framework for the relationship between sport and gratitude.

Another contribution of the study is to apply the developed instrument for the Social Impact of Sport of Lee, Cornwell, and Babiak's work [6], published in the Journal of Sport Management, into the real industry of a sport-based development initiative in Singapore. The findings of the study can help various stakeholders justify their investment and engagement in the sport-based national development initiative of Vision 2030 with both theoretical and empirical evidence. In addition, the findings from theoretical and empirical evidence are useful to build up a strategic map regarding how to better leverage the impact of sport on society. For example, it implies that how grateful participants/ beneficiaries feel during or through sport activities or events could be one of the most important factors to enhance the social outcomes in the sport-based development initiative.

The findings from the measurement model raise a challenge regarding why the instrument successfully captured only the two discriminant constructs of social outcomes, social capital and health literacy, not the five discriminant constructs as developed and tested in the work of Lee, Cornwell, and Babiak [6]. One of the reasons might come from the samples' characteristics and environmental differences in the interviewing processes. When the instrument was developed and tested with a sample of college students in a classroom setting, the students might be able to better understand the constructs and distinguish their similar question items of social capital, collective identities, health literacy, well-being, and human capital due to their active academic ability. However, when the instrument was applied with the general public (regardless of education level and age) in the real context of a sport-based development initiative, some question items (e.g., social capital vs. collective identity, well-being vs. human capital) can be similarly understood and perceived by respondents. It might be considered as a gap between academia and practical work in sport management. It also implies that the instrument of Lee, Cornwell, and Babiak [6] may need to be refined more specifically for its effective application into the sport industry with realties and complexities, depending on the characteristics of target populations and data collection environment. Additionally, another reason might come from the sample's English proficiency capacity. Although Singaporeans typically speak English, they are not native English speakers. When the instrument was tested with college students in US, they could easily respond to and distinguish the similar question items. On the other hand, it might not have been that easy for Singaporeans to respond to and distinguish the similar question items of social capital, collective identities, and human capital within the limited time of the interviewing process. This issue might have influenced the discriminant validity. We can also think that the possible reason might be related to who administered the survey (principal investigator vs. trained interviewers). When the principal investigator administered the survey from a classroom, the respondents were more likely to better understand the survey questions and pay more attentions to the survey processes. Additionally, the principal investigator could better control the survey process. On the other hand, when the interviewers administered the door-to-door survey, these conditions and the attitudes of the respondents might not be the same as when the principal investigator administered.

*Limitations and Future Research*

We have tested the conceptualized reciprocity mechanism of a sport-based initiative with a random sample of the general public of a city-state, considering the population as participants and beneficiaries at the same time in a sport-based national development initiative. This research still addresses significant relationships and outcomes in the reciprocity mechanism. Future research should apply this model into a more specific target population, especially those who are beneficiaries from corporate sponsorship,

corporate social responsibility, nonprofit partnerships, government subsidy programs (e.g., sports voucher), and athlete foundations. Given that gratitude is associated with individuals' welfare and various positive developmental outcomes for children and adolescents [53], it can examine particular aspects of gratitude with regard to beneficiaries, benefactors, and society at large in sport-based cause-oriented sponsorship and corporate social responsibility practices. Additionally, it can give us opportunities to develop and test a more integrated model of reciprocity mechanism by including various stakeholders in sport-based initiatives. Further, it gives us opportunities to examine fully the dynamic nature of gratitude to influence the economic or social objectives of stakeholders in sport-based initiatives. For example, if we strategically communicate that beneficiaries experience and receive important social benefits from a sport-based initiative sponsored by a corporation, it might be effective to reduce the criticized commercialism for corporate engagement in corporate social sponsorship and CSR [3].

Admittedly, hypothetical bias and social desirability bias cannot be completely excluded in measuring the prosocial behavior intention of willingness to donate (WTD) to societal foundations. Future study is required to examine the prosocial behavior of beneficiaries/participants in sport-based social initiatives in more comprehensive and longitudinal ways. For example, some beneficiaries (e.g., sick children, football scholarship students) might require time for their recovery, development, maturity, or success before they can be able to act more prosocially. Thus, longitudinal research is necessary to fully examine these underlying or time-required relationships of the reciprocity mechanism of sport-based social initiatives. We also admit that the data collection is slightly outdated, since the data were collected in 2014 and 2015. Future study is required to examine the model with the newest data collection to generalize the relationships and findings at present.

## 6. Conclusions

When we intend to accomplish sustaining reciprocity relationships in sport-based social initiatives, we face many challenges in the real world. The current research tried to address two of those issues by measuring the invisible outcomes in a more standardized way and by identifying gratitude as a mediator to enhance the social outcomes. Especially, the study developed an integrated conceptual model for the sustainable reciprocity relationships among sport participation, gratitude, social benefits, and prosocial behaviors in a sport-based initiative. Subsequently, the study empirically tested the model and its mechanism of structural relationships for participants and their prosocial behaviors in the context of a real sport-based national initiative. One of the key findings is that sport participation can lead to prosocial behaviors through the serial mediation of gratitude and the perceived social capital. It implies how significant the perceived value of gratitude and social capital value is for participants to enhance their prosocial behaviors in sport social initiatives. These research efforts will play a pivotal role in giving us a better understanding of how gratitude can play a key role in initiating a new paradigm of sport-based sponsorship, partnerships, and CSR practices. They will foster "real win–win—win" relationships among beneficiaries, benefactors, and society by examining the sustainable reciprocity mechanism of gratitude to influence or facilitate the outcomes for beneficiaries as well as spur beneficiaries, benefactors, and the general public to act more prosocially in sport-based social partnerships and CSR practices.

**Funding:** This research was funded by Hankuk University of Foreign Studies Research Fund 2018.

**Acknowledgments:** The author is particularly grateful to the editors and reviewers for their suggestions and comments on improving this study.

**Conflicts of Interest:** The author declares no conflict of interest.

## Appendix A. An Example of Measurement of Donation Behavioral Intention Using the Contingent Valuation Method (CVM)

The National Football Academy (NFA) was launched with the aim of developing Singapore's young footballers. NFA supports young footballers to play for Singapore in the international games for the glory of Singapore, representing Singapore as an ambassador. NFA also provides the young footballers with meaningful exposures to local communities of Singapore through the Lion City Cup (LCC). Families, friends, and colleagues enjoyed civic pride and community bonds, gathering and cheering on their heroes at the 23rd, 24th, and 25th LCC in June 2011, 2012, and 2013. Many fans agree that they were inspired by the positive image and excellent performance and that they experienced enhanced national pride and identity as Singaporean. Further, NFA builds up community programs for young footballers to participate in community outreach projects and make voluntary contributions to communities. These activities develop educational values, revealing new possibilities and transforming challenges and adversity into joy and success for children, youths, and communities through inspiration and role modelling from sporting heroes in Singapore.

Hypothetically suppose that NFA Foundation is established as a non-profit to ensure that NFA is properly managed to benefit communities, society and Singapore for the values as described: (1) civic pride, (2) community bonds, (3) national pride and identity, (4) inspiration and role modelling from sporting heroes. Without individual contributions, NFA Foundation cannot play its intended beneficial roles to society.

Based on the scenario above and considering all the positive benefits and contributions that NFA makes to Singaporean society, how much would you be willing to pay as a monthly donation to the NFA Foundation?

Please circle one answer for each amount level to show your certainty of willingness to donate.

| The Amount | Level of Your Certainty | | | | |
|---|---|---|---|---|---|
| S$ 1 | Definitely Yes | Probably Yes | Don't know | Probably No | Definitely No |
| S$ 2 | Definitely Yes | Probably Yes | Don't know | Probably No | Definitely No |
| S$ 5 | Definitely Yes | Probably Yes | Don't know | Probably No | Definitely No |
| S$ 10 | Definitely Yes | Probably Yes | Don't know | Probably No | Definitely No |
| S$ 20 | Definitely Yes | Probably Yes | Don't know | Probably No | Definitely No |
| S$ 50 | Definitely Yes | Probably Yes | Don't know | Probably No | Definitely No |
| S$ 100 | Definitely Yes | Probably Yes | Don't know | Probably No | Definitely No |
| S$ 200 | Definitely Yes | Probably Yes | Don't know | Probably No | Definitely No |
| S$ 500 | Definitely Yes | Probably Yes | Don't know | Probably No | Definitely No |
| S$ 1000 and over | Definitely Yes | Probably Yes | Don't know | Probably No | Definitely No |

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
