# Peer review of "Sustainable Reciprocity Mechanism of Social Initiatives in Sport: The Mediating Effect of Gratitude"

_sustainability, doi:10.3390/su12219279_

Round 1

Reviewer 1 Report

    It is viewed that this study could contribute to the existing literature about the role of sport participation in inducing positive social benefits and prosocial behaviors in sport-based social initiatives. In particular, this manuscript also offers empirical evidence that gratitude mediates the effects of sport participation on social benefits in sport-based social initiatives.

    While this manuscript has a potential to be considered for publication, there are several conceptual and measurement issues for further clarifications or explanations.

    First, as shown in the manuscript title, the conceptual model of “sustainable reciprocity mechanism” needs further detailed explanations such as what it means by the conceptual model and why it is important. Without discussing about solid theoretical groundings about the central conceptual model of “sustainable reciprocity”, the manuscript appears to assume probable relationships among sport participation, gratitude, social benefits, and prosocial behaviors in sport-based initiatives.

    Second, as the authors acknowledge, in the discussion section, that the measurement model demonstrates shortcomings in its structural composition of five constructs. For example, the discriminant validity of the proposed social benefit constructs (i.e., social capital, well-being, health literacy, collective identities, and human capital), developed and tested in the authors’ previous research, turn out to satisfy for only two constructs – social capital and health literacy. Although the authors provide one of the reasons for this significant issue of the measurement model, their explanation does not sound convincing. Then, authors’ discussion about this issue needs to include additional persuasive accounts.

    Third, with only two constructs used for actual analyses of the model, health literacy even shows no significant influence on the prosocial behavior measurement. However, the authors do not discuss reasons for the not-statistically-significant relationship that, then, does not support the hypothesis based on the conceptual model. Conceptual elaborations should be further discussed because it seems that health literacy indicates personal outcome rather than social benefits. Moreover, it can be conceived that this individualized conceptual nature of health literacy would result in the no significant statistical relationship with the prosocial behavior measurement.

Author Response

Response to Reviewer 1 comments

    It is viewed that this study could contribute to the existing literature about the role of sport participation in inducing positive social benefits and prosocial behaviors in sport-based social initiatives. In particular, this manuscript also offers empirical evidence that gratitude mediates the effects of sport participation on social benefits in sport-based social initiatives.

    While this manuscript has a potential to be considered for publication, there are several conceptual and measurement issues for further clarifications or explanations.

Author: Thank you for reviewing the manuscript and offering your insightful comments and suggestions for revision. I have tried to address all the issues that you mentioned. I think the manuscript is further developed. I really appreciate your review. 

 Point 1: First, as shown in the manuscript title, the conceptual model of “sustainable reciprocity mechanism” needs further detailed explanations such as what it means by the conceptual model and why it is important. Without discussing about solid theoretical groundings about the central conceptual model of “sustainable reciprocity”, the manuscript appears to assume probable relationships among sport participation, gratitude, social benefits, and prosocial behaviors in sport-based initiatives.

Response 1: I agree with your point. In order to integrate your comment here, I have provided more explanations regarding what the conceptual model means, why it is important in the Introduction and further what the significant contributions of the conceptual model are in the Discussion section. Please review the lines 41 to 53 and 68 to 71 of the revised manuscript in the introduction and the lines 434 to 448. Thank you for your insightful feedback.   

Point 2: Second, as the authors acknowledge, in the discussion section, that the measurement model demonstrates shortcomings in its structural composition of five constructs. For example, the discriminant validity of the proposed social benefit constructs (i.e., social capital, well-being, health literacy, collective identities, and human capital), developed and tested in the authors’ previous research, turn out to satisfy for only two constructs – social capital and health literacy. Although the authors provide one of the reasons for this significant issue of the measurement model, their explanation does not sound convincing. Then, authors’ discussion about this issue needs to include additional persuasive accounts.

Response 2: Thank you for your critical feedback on it. I have provided more explanations on this issue in the Discussion section. Please review the lines 489-501.

Point 3: Third, with only two constructs used for actual analyses of the model, health literacy even shows no significant influence on the prosocial behavior measurement. However, the authors do not discuss reasons for the not-statistically-significant relationship that, then, does not support the hypothesis based on the conceptual model. Conceptual elaborations should be further discussed because it seems that health literacy indicates personal outcome rather than social benefits. Moreover, it can be conceived that this individualized conceptual nature of health literacy would result in the no significant statistical relationship with the prosocial behavior measurement.

Response 3: Thank you for your comment and your insightful explanation. I have tried to address this issue, but I think your explanation perfectly fit in the discussion. Would you please allow me to use your wordings in the manuscript? Please review the lines 449-455. It would be greatly appreciated. Thank you for you all your feedback, comments and ideas to refine the manuscript.

Reviewer 2 Report

First of all, I would like to congratulate the author for this wonderful work. It shows the hard-working of the author.

The present study aimed to investigate the conceptual model for sustainable reciprocity relationships among sport participation. Almost all of the sections are well performed. The results are reasonable with the acceptable model fit. However, there are still some points that may need to be clearly addressed or can be improved.

  1. Some sentences might be too wordy and hard to be read.
  2. It seems some imbalances have existed in the study population, in particular the race. Especially the forms of gratitude are more or less different among cultures and ethnicities. It might create deviation and need to be concerned.
  3. Although the results showed the good model fit, the items within gratitude seem to be too similar for the participants. I suggest the author re-check and depict the correlation among the items. It might have no discrimination among the items if the correlation is too high.
  4. The present study tries to have a better understanding of the sustainable mechanisms on sport participation. However, it seems less discussed from the view of motivation, self-determination, or self-efficacy. These theories are also essential and supported the results. I suggest the author discuss the result from a sport psychology perspective to provide a more informative and reasonable discussion. 

Author Response

Response to Reviewer 2 comments

First of all, I would like to congratulate the author for this wonderful work. It shows the hard-working of the author.

The present study aimed to investigate the conceptual model for sustainable reciprocity relationships among sport participation. Almost all of the sections are well performed. The results are reasonable with the acceptable model fit. However, there are still some points that may need to be clearly addressed or can be improved.

Author: Thank you for reviewing the manuscript and offering your insightful comments and suggestions for revision. I have tried to address all the issues that you mentioned. I think the manuscript is further developed. I really appreciate your review.

Point 1: Some sentences might be too wordy and hard to be read.

Response 1: Thank you for your comment. I have tried to work on this issue especially in the Introduction section and Discussion section. 

Point 2: It seems some imbalances have existed in the study population, in particular the race. Especially the forms of gratitude are more or less different among cultures and ethnicities. It might create deviation and need to be concerned.

Response 2: I totally agree with your point. But, the current sample was chosen to be representative of the Singaporean population in terms of race, sex, income, education level. So, it has both strengths and weakness. When we keep balance in terms of race, it does not represent the Singaporean population. So, I hope you understand this point. Thank you.   

Point 3: Although the results showed the good model fit, the items within gratitude seem to be too similar for the participants. I suggest the author re-check and depict the correlation among the items. It might have no discrimination among the items if the correlation is too high.

Response 3: Thank you for your comment. I have check the items within gratitude again and found the result is correct. Also, I have included Table 4 in the line 27, which shows correlations among the constructs of social outcomes and gratitude. It confirms discriminant validity among gratitude, social capital and health literacy. Thank you for your critical comment.

Point 4: The present study tries to have a better understanding of the sustainable mechanisms on sport participation. However, it seems less discussed from the view of motivation, self-determination, or self-efficacy. These theories are also essential and supported the results. I suggest the author discuss the result from a sport psychology perspective to provide a more informative and reasonable discussion. 

Response 4: Thank you for your insightful comment. I have included more discussions based on the previous sport psychology literature. Please review the lines 432-452. Thank you for your suggestion.

Reviewer 3 Report

First of all, I appreciate that you have sent me this manuscript for review. The subject is interesting, but there are limitations in the study that the authors should review and include possible improvements:

In title section:

Despite the fact that the current title includes all the key concepts, it could be reduced. “Investigating the” is removable.

In abstract section:

Authors could think about the possibility of including one first sentence about the theoretical framework that embraces this paper.

The simple size of each study should be indicated.

There is a spelling mistake when it is said “The findings provides” instead of “The findings provide”.

In keywords section:

This reviewer suggests that keywords should not be repeated in the title (gratitude) so as to increase the visibility of the paper in different databases.

In introduction section:

The used references are, most of them, easy to be updated. Authors should try to change the references into new documents published, when possible, in 2019 or 2020. Besides, the references style is really poor done. A serious review with this regard is needed.

When a quote is literal (lines 44-48) and exceed 40 words, the margins should change.

The sentence in lines 75-77 could be with a higher range of references [7-16]. For instance, the literature published about Personal and Social responsibility model in physical education is a constantly growing field of research. Specifically, in Spain there are many researchers publishing about this topic.

Authors could consider the possibility of presenting the five core areas to which SEPE can make a social contribution by Lawson (2005) (lines 85-96) in Table 1.

Since the data collection was carried out in 2014 and 2015, its information is a bit outdated. This fact should be considered as a limitation.

In Study 2 section:

The sample size is said to be n = 500 (line 328) and n = 501 (line 333). Please, correct the wrong data.

In discussion section:

This section is really important since it should analyze the advances that this paper aims to generate. Nevertheless, the current discussion is far from achieving it. There is only one quote (Lee, Cornwell and Babiak, 2013). The results are not tried to explained through the already published literature. Moreover, their implications are neither enough developed nor justified.

Authors should include a final section called Conclusions, where they explained what they have achieved with regard to their goals.

In references section:

This section is, currently, very weak. There is a huge amount of mistakes since hardly any differences are well written according to Sustainability references norms.

Author Response

Respose to Reviewer 3 comments

First of all, I appreciate that you have sent me this manuscript for review. The subject is interesting, but there are limitations in the study that the authors should review and include possible improvements:

Author: Thank you for reviewing the manuscript and offering your insightful comments and suggestions for revision. I have tried to address all the issues that you mentioned. I think the manuscript is further developed. I really appreciate your review.

In title section:

Point 1: Despite the fact that the current title includes all the key concepts, it could be reduced. “Investigating the” is removable.

Response 1: Thank you for your suggestion. It has been removed.

In abstract section:

Point 2: Authors could think about the possibility of including one first sentence about the theoretical framework that embraces this paper.

Response 2: Thank you for your suggestion. Yes, it has been included in the abstract.

Point 3: The simple size of each study should be indicated.

Response 3: Thank you for your suggestion. It has been included in the abstract.

Point 4: There is a spelling mistake when it is said “The findings provides” instead of “The findings provide”.

Response 4: Thank you for your point. It has been corrected.

In keywords section:

Point 5: This reviewer suggests that keywords should not be repeated in the title (gratitude) so as to increase the visibility of the paper in different databases.

Response 5: Thank you for your suggestion. It has been changed.

In introduction section:

Point 6: The used references are, most of them, easy to be updated. Authors should try to change the references into new documents published, when possible, in 2019 or 2020. Besides, the references style is really poor done. A serious review with this regard is needed.

Response 6: Thank you for your suggestion. The references have been updated.

Point 7: When a quote is literal (lines 44-48) and exceed 40 words, the margins should change.

Response 7: Thank you for your critical comment. This issue has been addressed. Please review the lines 49-53.

Point 8: The sentence in lines 75-77 could be with a higher range of references [7-16]. For instance, the literature published about Personal and Social responsibility model in physical education is a constantly growing field of research. Specifically, in Spain there are many researchers publishing about this topic.

Response 8: Thank you for your suggestion. The recent references have been included. Please review the reference list [7-20].

Point 9: Authors could consider the possibility of presenting the five core areas to which SEPE can make a social contribution by Lawson (2005) (lines 85-96) in Table 1.

Response 9: Thank you for your suggestion. But, I think it is appropriate to present the definitions for the measurement variable with readers in Table 1 rather than the five core areas suggested by Lawson (2005).  

Point 10: Since the data collection was carried out in 2014 and 2015, its information is a bit outdated. This fact should be considered as a limitation.

Response 10: Thank you for your comment. This issue has been discussed as a limitation. Please review the lines 529-531.

In Study 2 section:

Point 11: The sample size is said to be n = 500 (line 328) and n = 501 (line 333). Please, correct the wrong data.

Response 11: Thank you for your point. It has been corrected.

In discussion section:

Point 12: This section is really important since it should analyze the advances that this paper aims to generate. Nevertheless, the current discussion is far from achieving it. There is only one quote (Lee, Cornwell and Babiak, 2013). The results are not tried to explained through the already published literature. Moreover, their implications are neither enough developed nor justified.

Point 12: Thank you for your insightful comment. I have included more discussions based on the previous sport psychology literature. Please review the lines 432-452. Thank you for your suggestion.

Point 13: Authors should include a final section called Conclusions, where they explained what they have achieved with regard to their goals.

Point 13: Thank you for your suggestion. Now the conclusion is included. Please review the lines 533-550.

In references section:

Point 14: This section is, currently, very weak. There is a huge amount of mistakes since hardly any differences are well written according to Sustainability references norms.

Point 14: Thank you for your critical comment. I have corrected the references to follow the format of the journal.

Reviewer 4 Report

The manuscript entitled Investigating the Sustainable Reciprocity Mechanism of Social Initiatives in Sport: The Mediating effect of Gratitude is considered original. No plagiarism detection in the manuscript. All requirements by the reviewers are included and carefully conducted. The paper presented two studies that are required more synergy in the conclusion. The first objective of the study is to explore a conceptual model for the sustainable reciprocity relationships among sport participation, gratitude, social benefits, and prosocial behaviors in a sport-based initiative based on previous literature on the social impact of sport and gratitude. The second objective is to present the social outcomes, gratitude and prosocial behaviours. Those objectives are very organized in the paper in different sections. However the conclusion is under the expectations. The two cases presented are well prepare but are required a more strong and clear conclusions in the manuscript.  In the section 3.2.2 it is suggested to include one figure for better understand of the Structural equation modeling (SEM). The figure 2 can be improved related to the visual presentation, the font is not tottaly clear. The last suggestion is to present separetely, rewriting the "Limitations" of the study and move this item to after the methodology. 

Author Response

Response to Reviewer 4 comments

Comments and Suggestions for Authors

The manuscript entitled Investigating the Sustainable Reciprocity Mechanism of Social Initiatives in Sport: The Mediating effect of Gratitude is considered original. No plagiarism detection in the manuscript. All requirements by the reviewers are included and carefully conducted. The paper presented two studies that are required more synergy in the conclusion. The first objective of the study is to explore a conceptual model for the sustainable reciprocity relationships among sport participation, gratitude, social benefits, and prosocial behaviors in a sport-based initiative based on previous literature on the social impact of sport and gratitude. The second objective is to present the social outcomes, gratitude and prosocial behaviours. Those objectives are very organized in the paper in different sections.

Author: Thank you for reviewing the manuscript and offering your insightful comments and suggestions for revision. I have tried to address all the issues that you mentioned. I think the manuscript is further developed. I really appreciate your review.

Point 1: However the conclusion is under the expectations. The two cases presented are well prepare but are required a more strong and clear conclusions in the manuscript.

Response 1: Thank you for your comment. The conclusion has been included. Please review the lines 533-550.

Point 2: In the section 3.2.2 it is suggested to include one figure for better understand of the Structural equation modeling (SEM). The figure 2 can be improved related to the visual presentation, the font is not tottaly clear.

Response 2: Thank you for your comment. The figure for two structural equation models is separated and inserted as Figure 2 and Figure 3 accordingly. Now the font also is clear.

Point 3: The last suggestion is to present separetely, rewriting the "Limitations" of the study and move this item to after the methodology.

Response 3: Thank you for your comment. But, I think that the limitations of the study typically come after the sections of results and discussion in the articles. Also, it is more natural to admit the limitations of the study and subsequently discuss further research directions.   

Round 2

Reviewer 1 Report

It is clearly demonstrated that the authors have addressed my early concerns in the revised manuscript and I have found the revised manuscript substantially improved.

I think the manuscript has potential for making contribution to the field, and I will be happy to see it published.

Regards

Reviewer 2 Report

Great job! I can see the revision of this study has improved.

Reviewer 3 Report

Dear author,

after a second revision of your paper, I only have to congratulate you for its final state.